# The Molecular Mechanism of *GhbHLH121* in Response to Iron Deficiency in Cotton Seedlings

**DOI:** 10.3390/plants12101955

**Published:** 2023-05-11

**Authors:** Jie Li, Ke Nie, Luyao Wang, Yongyan Zhao, Mingnan Qu, Donglei Yang, Xueying Guan

**Affiliations:** 1State Key Laboratory of Crop Genetics and Germplasm Enhancement, Hybrid Cotton R & D Engineering Research Center (the Ministry of Education), College of Agriculture, Nanjing Agricultural University, Nanjing 210095, China; 18757598197@163.com; 2Zhejiang Provincial Key Laboratory of Crop Genetic Resources, Institute of Crop Science, Plant Precision Breeding Academy, College of Agriculture and Biotechnology, Zhejiang University, Hangzhou 300058, China; 12116024@zju.edu.cn (K.N.); 2171100066@zju.edu.cn (L.W.); 22016144@zju.edu.cn (Y.Z.); 3Hainan Institute, Zhejiang University, Yongyou Industry Park, Yazhou Bay Sci-Tech City, Sanya 572000, China; 4Hainan Yazhou Bay Seed Lab, Yazhou Bay Science and Technology City, Yazhou District, Sanya 572025, China; qmn@yazhoulab.com

**Keywords:** *GhbHLH121*, cotton seedling, iron deficiency, photosynthetic efficiency

## Abstract

Iron deficiency caused by high pH of saline–alkali soil is a major source of abiotic stress affecting plant growth. However, the molecular mechanism underlying the iron deficiency response in cotton (*Gossypium hirsutum*) is poorly understood. In this study, we investigated the impacts of iron deficiency at the cotton seedling stage and elucidated the corresponding molecular regulation network, which centered on a hub gene *GhbHLH121*. Iron deficiency induced the expression of genes with roles in the response to iron deficiency, especially *GhbHLH121*. The suppression of *GhbHLH121* with virus-induced gene silence technology reduced seedlings’ tolerance to iron deficiency, with low photosynthetic efficiency and severe damage to the structure of the chloroplast. Contrarily, ectopic expression of *GhbHLH121* in *Arabidopsis* enhanced tolerance to iron deficiency. Further analysis of protein/protein interactions revealed that GhbHLH121 can interact with GhbHLH IVc and GhPYE. In addition, GhbHLH121 can directly activate the expression of *GhbHLH38*, *GhFIT,* and *GhPYE* independent of GhbHLH IVc. All told, GhbHLH121 is a positive regulator of the response to iron deficiency in cotton, directly regulating iron uptake as the upstream gene of *GhFIT*. Our results provide insight into the complex network of the iron deficiency response in cotton.

## 1. Introduction

With the worldwide decrease in available arable land, the efficient use of soil has become an important research direction. About 30% of the world’s arable land is saline–alkali soil with pH above 8.2 [1,2,3]. Saline soil refers to sodium salt soil mainly containing NaCl, while alkaline soil mainly refers to that containing Na_2_CO_3_ and NaHCO_3_ [4,5]. Alkaline soil means soil with a high pH value. When soil pH is above 7.4, the solubility of iron hydroxide is decreased to 10^−18^ M, at which level it cannot be effectively absorbed and utilized by plants [6]. Iron is an essential microelement for plant growth and development, playing important roles in the electron transport chain (ETC) and in enzymatic reactions that contribute to many physiological metabolic processes such as photosynthesis, respiration, nitrogen fixation, and protein as well as nucleic acid synthesis [7,8,9]. Iron deficiency inhibits plant growth and development, evidenced in leaf chlorosis with decreased chlorophyll content, plant biomass, iron content, and increased iron reductase activity [10,11,12]. Cotton is a global cash crop with higher saline–alkali tolerance than other food crops [1,13]. However, when the alkaline level in the soil is too high, the cotton seedling will still be contaminated by the alkaline levels, characterized by low seedling germination rate, slow growing development, and high death rate [14,15]. In addition, iron-deficient cotton presents with chlorosis, thicker roots, and undeveloped root hairs [16,17]. Therefore, it is essential to improve the utilization rate of iron and enhance tolerance to iron deficiency at the cotton seedling stage. However, at present, the molecular regulation of iron deficiency in the cotton genome is barely understood.

About 60% of the iron in plant leaves is fixed in the thylakoid membrane and matrix of chloroplasts [9,18], where its ability to donate and accept electrons plays a significant role in electron transfer reactions. Iron is present in all electron transfer complexes, PSI, PSII, cytochrome b6f complexes, and ferredoxins, and is also indispensable for the biogenesis of cofactors such as hemes and iron sulfur clusters [19]. Iron deficiency can impede photosynthesis and introduce chaos into the chloroplast ultrastructure [20], as has been observed in maize [21] and rice [22], with decreased starch grains and an accumulation of lipid in plastid globules.

To improve iron absorption and utilization, plants have evolved two main strategies for iron uptake: the reduction strategy (strategy I) for dicotyledons and non-gramineous monocotyledons, and the chelation strategy (strategy II) for gramineous monocotyledons [8,9,23,24,25]. *Arabidopsis*, which utilizes strategy I, employs H+-ATPase 2 (AHA2) to release protons into the soil, thereby reducing soil pH and improving Fe solubility [26]. Then, the Fe chelate reductase FERRIC REDUCTION OXIDASE 2 (FRO2) on the root surface catalyzes the reduction in Fe^3+^ to Fe^2+^, which is a key step in iron absorption [27]. Iron-regulated Transporter 1 (IRT1) then transports the Fe^2+^ into root cells [28,29,30,31,32]. The chelation strategy of gramineous plants (strategy II) involves the secretion of phytosiderophores (PS) such as mugineic acid (MAS) into the rhizosphere to form Fe^3+^-Ps chelates, which are then transported into the plant by Yellow stripe1/Yellow Stripe1-like (YSL) family transporters [33,34].

Unsurprisingly, plants have developed a series of complex regulatory systems to maintain iron homeostasis at both the transcriptional and post-transcriptional levels. Three regulatory pathways have been described, with respective hub genes fer-like iron deficiency-induced transcription factor (*FIT*) [35,36], POPEYE (*PYE*) [37,38], and BRUTUS (*BTS*) [39,40,41]. In *Arabidopsis*, FIT can form a complex with bHLH Ib to directly regulate the iron uptake genes *AHA2, FRO2*, and *IRT1*, which directly participate in the iron uptake process as part of the reduction mechanism [42]. PYE ensures iron redistribution in plants by regulating *YSL1*, *NAS4,* and *FRD3* [37]. *PYE*, *bHLH11* [43,44], and *bHLH121* [45,46,47] all belong to the same subgroup, bHLH IVb. *PYE* and bHLH11 have been reported as negative regulators of iron homeostasis [43,44,48]. However, *bHLH121* is a positive regulator with a controversial expression pattern under iron deficiency in various studies [45,46,47]. Iron deficiency also promotes the accumulation of phosphorylated bHLH121, which activates the transcription of downstream target genes including bHLH Ib TFs and *FIT*, but only when bHLH121 binds to bHLH IVc transcription factors (TFs) [46,47]. These results indicate that bHLH121 functions as an upstream regulator of iron homeostasis [23,49,50,51].

Cotton is an important source of renewable fiber for the textile industry, and the cotton seedling stage suffers from iron deficiency on saline–alkali land. Most studies of saline–alkali land in cotton have focused on salt stress. In recent years, more studies have begun to focus on the alkali stress in saline–alkali land; an in vitro culture assay demonstrated that iron deficiency prevented cotton fiber development and was associated with decreased ferric reduction activity as well as increased activation of *GhFRO2*, *GhIRT1*, *GhFIT1*, and *GhILR3* [52], which suggested cotton may adapt strategy I in its response to iron deficiency. Nonetheless, the specific regulatory mechanism was still not clear. This study investigated the regulation of iron homeostasis in cotton. This study provides new genetic resources and a theoretical basis for developing new cotton germplasms with improved tolerance to iron deficiency.

## 2. Results

### 2.1. Iron Deficiency Induces Differential Expression of Genes Related to Iron Deficiency with Consequent Low Photosynthetic Efficiency in Cotton

Fe has a very low solubility in saline–alkali soil with high HCO_3_^-^ concentrations, resulting in limited uptake by plant roots [4]. Iron deficiency caused cotton leaf chlorosis, thicker roots, and fewer root hairs (Figure 1A) [16,17]. In order to study the regulation mechanism governing the cotton seedling stage response to iron deficiency, we first irrigated the laboratory-grown cotton seedlings with ^1^/_2_ Hoagland nutrient solution either with (+Fe) or without Fe^2+^ (−Fe). Under −Fe conditions, for two weeks, cotton seedlings appeared stunted and had chlorotic leaves (Figure 1B). Leaf chlorophyll content (Figure 1C), leaf biomass (Figure 1D), Fe content (Figure 1E), and net photosynthesis rate (Figure 1G, Appendix A) were significantly lower in the −Fe group than those in the control group, while ferric chelate reductase activity (FCR) was increased (Figure 1F). The photosynthetic electron transport machinery is composed of two light energy-driven photosystems (PSs), PSI and PSII, along with the cytochrome (Cyt) *b_6_f* and ATP synthase. In linear electron transport (LET), electrons extracted from water by PSII were transported to PSI through Cyt*b_6_f* and eventually produced NADPH [53,54]. The iron deficiency treatment suppressed expression of genes encoding proteins involved with PSI, PSII, Cyt*b_6_f*, and Ferredoxin (Fd) (Appendix A). Iron deficiency also affected the accumulation of reactive oxygen species (ROS) in cotton leaves (Appendix A). Staining with nitroblue tetrazolium (NBT) (Appendix A) and diaminobenzidine tetrahydrochloride (DAB) (Appendix A) alike was darker on leaves of −Fe plants than those from +Fe control plants. These results indicate that the content of O_2_- and H_2_O_2_ on cotton leaves is increased under −Fe conditions. Examination of superoxide dismutase (SOD), catalase (CAT), and peroxidase (POD) revealed the −Fe group exhibiting higher activities of SOD (Appendix A) and POD (Appendix A) in cotton leaves compared with the control group. CAT content did not differ significantly (Appendix A). All told, cotton grown under iron deficiency conditions showed low photosynthetic efficiency.

In order to study transcriptional regulation in iron-deficient cotton, a preliminary analysis was carried out of the expression of genes that may be involved in the response to iron deficiency. Firstly, the amino acid sequences of genes known to be involved in iron transport and iron signal regulation in *Arabidopsis* were extracted, and homologous genes in the cotton genome were identified by construction of a phylogenetic tree (Appendix A). Cotton seedlings were then grown for two weeks with +Fe or −Fe treatment and the leaves harvested for reverse transcription qRT-PCR analysis. Expression of *GhbHLH121*, *GhPYE* (bHLH IVb), and *GhbHLH104* (bHLH IVc) was found to be increased in the −Fe group, while *GhbHLH115* (bHLH IVc) did not show a difference. Members of the bHLH Ib subfamily which functioned in a FIT-dependent transcriptional regulatory pathway were differentially impacted, *GhbHLH38* was elevated significantly in the −Fe group, but another subfamily member, *GhbHLH101*, did not show a significant difference. *GhFIT* (bHLH IIIa) and *GhBTS*, which encoded a member of the protease degradation pathway, were both increased under −Fe conditions. The FIT-dependent iron-uptake genes *GhIRT1, GhFRO2*, and *GhAHA2* also showed significantly higher expression in the −Fe group (Figure 1H). These results indicate that iron deficiency is a substantial stressor for cotton seedlings and has direct impacts on the transcription of iron uptake- and homeostasis-related genes.

### 2.2. GhbHLH121 Encodes a Transcription Factor Involved in Responding to Iron Deficiency in Root and Shoot Tissues

Iron deficiency induces the expression of *GhbHLH121*, which encodes a homolog of AtbHLH121 (86.62% amino acid similarity), an upstream regulatory TF for iron homeostasis in *Arabidopsis* (Appendix A). The specific function of *GhbHLH121* in the cotton has not been characterized. *GhbHLH121* encodes a protein of the bHLH IVb subfamily that features a canonical bHLH domain (Appendix A). The TM-1 reference genome for allotetraploid upland cotton features two copies of *GhbHLH121* from the A and D subgenomes which have 97.44% similarity in their DNA sequences (Appendix A). Transcriptome data show that *GhbHLH121-A_T_* and *GhbHLH121-D_T_* are highly expressed in roots and stems, followed by leaves, but expressed at lower levels in petals and sepals of floral organs. The expression patterns of GhbHLH121-AT and GhbHLH121-DT were not significantly different because they were more expressed in roots and leaves than in floral organs (Figure 2A). qRT-PCR further showed that *GhbHLH121* expression predominantly occurred in roots and leaves at the seedling stage and continued to increase as leaves grew (Figure 2B), with minimal levels in floral organs. In addition, the expression of *GhbHLH121* in roots and leaves after −Fe treatment was significantly higher than that in the control (Figure 2C,D). The subcellular localization signals of GhbHLH121-A_T_ and GhbHLH121-D_T_ were distributed in the nucleus and cytoplasm (Figure 2E). Taken together, these findings indicate that *GhbHLH121* is an iron deficiency-induced gene expressed in both roots and leaves. The homeologs of *GhbHLH121-A_T_* and *GhbHLH121-D_T_* are not different in terms of coding sequence, transcriptional activity, or protein localization. Therefore, the gene name will not be distinguished in subsequent experiments and is written as *GhbHLH121* hereafter.

### 2.3. Suppression of GhbHLH121 in Cotton Seedlings Reduced Tolerance to Iron Deficiency

In order to analyze the role of *GhbHLH121* in regulating iron deficiency in cotton, we performed an *Agrobacterium*-mediated VIGS assay (virus-induced gene silence) to suppress *GhbHLH121* expression in cotton seedlings. The VIGS-treated seedlings were incubated in hydroponic culture with ^1^/_2_ Hoagland nutrient solution either with (+Fe) or without Fe^2+^ (−Fe) for three weeks. Under −Fe conditions, *TRV2:00* plants showed the obvious corresponding phenotype, that is, leaf chlorosis (Figure 3A, Appendix A). The expression of *GhbHLH121* was confirmed to be suppressed in *TRV2:GhbHLH121* plants relative to *TRV2:00* plants (Figure 3B) under both +Fe and −Fe conditions. Overall, under the −Fe conditions, *TRV2:00* and *TRV2:GhbHLH121* both showed decreased chlorophyll and Fe content, and this resulted in less leaf biomass (Figure 3C,F,G). Under the +Fe conditions, suppression of *GhbHLH121* led to slightly but significantly less biomass and Fe content than the control (Figure 3C,G). In the −Fe conditions, suppression of *GhbHLH121* greatly decreased chlorophyll and Fe content, leaf biomass, and FCR relative to *TRV2:00* plants (Figure 3C–G). The concomitant suppression of *GhbHLH121* further decreased the net photosynthetic rate (Figure 3E, Appendix A) and also the expression of genes encoding proteins involved with PSI, PSII, Cyt*b_6_f*, and Fd (Appendix A). In contrast, SOD activity in *TRV2:GhbHLH121* plants was significantly higher than that in *TRV2:00* under the −Fe conditions, though there was no significant difference in POD and CAT activities (Appendix A). Suppression of *GhbHLH121* thus reinforced the damage from iron deficiency, indicating that *GhbHLH121* should be a positive regulator of the response to iron deficiency. 

A further transcriptional examination of genes involved in iron uptake and iron signaling was conducted by using the material of the *TRV2:GhbHLH121* and *TRV2:00* (Table 1). qRT-PCR further showed that suppression of *GhbHLH121* did not affect the expression of *GhbHLH101* and *GhbHLH38* (bHLH Ib), nor that of *GhIRT1* and *GhFRO2* under the +Fe conditions. However, under −Fe conditions, *GhbHLH38*, *GhIRT1*, and *GhFRO2* all were significantly lower in *TRV2:GhbHLH121* than in *TRV2:00* controls, while *GhbHLH101* was not differentially expressed. In addition, *GhFIT* was significantly lower in *TRV2:GhbHLH121* than in *TRV2:00* under both +Fe and −Fe conditions. Meanwhile, expression of the iron transport gene *GhYSL1* decreased under −Fe, but that of *GhNAS4*, *GhZIF1*, and *GhFRD3* was increased. However, there was no significant difference in the expression of *GhZIF1* and *GhFRD3* between *TRV2:GhbHLH121* and *TRV2:00* plants. *GhYSL1* and *GhNAS4* were decreased in *TRV2:GhbHLH121* under −Fe conditions. It has been reported that PYE regulates *YSL1*, *NAS4*, and *FRD3* to facilitate iron redistribution in plants, while BTS regulates iron homeostasis by affecting the stability of bHLH IVc proteins [40,55]. This study found both *GhPYE* and *GhBTS* to be induced under −Fe conditions. Furthermore, in +Fe conditions, *GhPYE* showed similar expression in *TRV2:GhbHLH121* and *TRV2:00* plants, but *GhBTS* was decreased with *TRV2:GhbHLH121*. Meanwhile, in −Fe conditions, *GhPYE* and *GhBTS* exhibited significantly lower expression in *TRV2:GhbHLH121* than in *TRV2:00*. These findings support that *GhbHLH121* is actively involved in responding to iron deficiency in cotton through regulating iron uptake and iron signaling under iron deficiency.

In order to study the effects of iron deficiency stress on the submicroscopic structure of cotton leaves, we observed the leaves of *TRV2:GhbHLH121* and *TRV2:00* plants treated with +Fe or −Fe by transmission electron microscopy (TEM). In +Fe conditions, both *TRV2:GhbHLH121* and *TRV2:00* exhibited normal chloroplast morphology with the thylakoids well piled up (Figure 4A). The chloroplasts contained oval starch granules and a few plastid globules (PGs). The mitochondrial was also in a round shape with the inner ridges arranged in an orderly manner (Figure 4B). The nuclear double membrane and the cytoplasmic membrane structure were clearly observed (Figure 4C). Under −Fe conditions, however, submicroscopic observation of *TRV2:00* leaf cells revealed obvious damage to the morphology of organelles. The internal structure of chloroplasts was disordered with loose arrangements of thylakoids (Figure 4A). Chloroplast sizes were significantly different with the width decreased (Figure 4A; Appendix A), aspect ratio increased (Appendix A), and PG number increased (Appendix A). Mitochondrial structure was likewise damaged with the loss of inner ridges (Figure 4B) and increased aspect ratio (Appendix A). The nucleus overall remained a normal shape (Figure 4C), but the cell membrane was also damaged or broken (Figure 4C). Thus, iron deficiency leads to chloroplast thylakoid structural disorder in cotton leaves, and the suppression of *GhbHLH121* increases the damages associated with iron deficiency, which confirms it as playing a positive role in iron homeostasis. With suppression of *GhbHLH121*, the chloroplast structure was damaged, the membrane lipid peroxidation was aggravated, and ROS accumulated; the photosynthetic electron transport system was destroyed, which resulted in the decrease in the photosynthetic rate.

### 2.4. Ectopic Expression of GhbHLH121 in Arabidopsis Confers Enhanced Tolerance to Iron Deficiency 

To investigate the role of *GhbHLH121* in the regulation of iron homeostasis, transgenic *Arabidopsis* lines were generated through ectopically expressed *GhbHLH121* (*GhbHLH121-OE*) fused to the cauliflower mosaic virus (CaMV) 35S promoter. Under +Fe conditions, there was no significant difference between the *Arabidopsis* seedlings of *GhbHLH121-OE* and WT in terms of growth state, including root length, chlorophyll content, FCR, and Fe content. Under −Fe conditions, WT seedlings exhibited a pronounced iron deficiency phenotype, including inhibited root growth, reduced FCR, and Fe content (Figure 5A–E). Compared with WT, *GhbHLH121-OE* seedlings showed amelioration of iron deficiency symptoms, that is, the root lengths were twice that of WT (Figure 5A,C) and FCR and Fe contents were higher (Figure 5D,E). These results suggest that the ectopic expression of *GhbHLH121* can reduce iron deficiency damages to *Arabidopsis*. 

Furthermore, we analyzed whether the tolerance of *GhbHLH121-OE* plants to iron deficiency is related to the expression of iron-related genes by qRT-PCR (Table 2, Appendix A). In the transgenic lines evaluated here, expression of *GhbHLH121* (Figure 5B, Appendix A) was increased under −Fe. In addition, the iron uptake genes *IRT1* and *FRO2* were upregulated together with *FIT* and *bHLHIb* (Table 2). Under +Fe conditions, ectopic expression of *GhbHLH121* did not affect the expression of *FIT, bHLHIb, IRT1*, and *FRO2*. In addition, expression of *PYE* was significantly higher in *GhbHLH121-OE* than in WT under −Fe conditions, but the expression of *YSL1, NAS4*, and *FRD3* did not differ significantly. Expression of *BTS* was higher in *GbhHLH121-OE* than in WT under −Fe conditions. While bHLH IVc itself is not activated by iron deficiency and is not affected by *GhbHLH121* under either −Fe or +Fe conditions, the observed expression patterns suggest that *GhbHLH121* may effectively alleviate the symptoms of iron deficiency in *Arabidopsis* by regulating genes related to iron homeostasis.

### 2.5. GhbHLH121 Interacts with GhbHLH IVc Genes and GhPYE

In order to study the molecular regulatory network of *GhbHLH121* in cotton, *GhbHLH121* was inserted into *pGBKT7* to construct a bait vector, which was used to screen a library of cotton proteins via the yeast two-hybrid system. A total of 85 genes whose products potentially interact with GhbHLH121 were screened from the yeast library (Appendix A), including GhbHLH104, GhbHLH115 (bHLH IVc TF), and PYE (bHLH IVb TFs). Yeast two-hybrid assay and Luciferase complementation assays confirmed that GhbHLH121 can interact with GhbHLH104, GhbHLH115, and PYE (Figure 6A,B). The bimolecular fluorescence complementation technology (BIFC) further confirmed these interactions (Figure 6D). In addition, YFP fluorescence signals were observed to be localized in the nucleus for three fusion proteins (Figure 6C). However, Figure 2E shows that the subcellular localization signals of GhbHLH121 were distributed in the nucleus and cytoplasm. Figure 6D shows that heterodimer of GhbHLH121 and GhbHLH104, GhbHLH115, PYE resides in the nucleus.

### 2.6. GhbHLH121 Directly Activates Transcription of GhbHLH38, GhFIT, and GhPYE Independent of GhbHLH104 and GhbHLH115

bHLH transcription factors regulate the expression of target genes by recognizing and binding E-box DNA motifs [56,57]. Multiple E-boxes are present in the promoters of *GhbHLH38*, *GhFIT*, and *GhPYE* (Figure 7A; Appendix A). In order to explore whether GhbHLH121 can bind to these E-box motifs, we generated constructs with the E-box sites mutated with CA to TC (mE-box) [58]. Yeast one-hybrid assay confirmed GhbHLH121 could bind to E-box motifs on the promoters of *GhbHLH38*, *GhFIT*, and *GhPYE* (Figure 7B). Mutation of the E-boxes in each promoter decreased this binding efficiency (Figure 7D).

To further verify whether GhbHLH121 has transcriptional activation effects on *GhbHLH38*, *GhFIT*, and *GhPYE*, the promoters of *GhbHLH38*, *GhFIT*, and *GhPYE* were fused into the LUC vector as reporter genes, for which GhbHLH121 was used as the effector (Figure 7C). Co-injection of constructs into tobacco leaves revealed that GhbHLH121 can activate the transcription of *GhbHLH38*, *GhFIT*, and *GhPYE* (Figure 7D,E). 

Lei et al. reported AtbHLH121 alone had no notable effect on FIT and bHLH38 promoter activity, while AtbHLH104 and AtbHLH115 each induced significant activation [46]. In addition, bHLH IVc TFs were reported to directly bind the bHLH121 promoter [46]. Therefore, we designed the experiment to explore whether the regulatory networks of cotton and Arabidopsis thaliana are consistent. GhbHLH104 and GhbHLH115 were used as effectors and the *GhbHLH121* promoter in the reporter (Figure 7F). GhbHLH104 and GhbHLH115 are unable to activate the transcription of *GhbHLH121* (Figure 7G,H). Thus, GhbHLH121 can activate transcription of *GhbHLH38*, *GhFIT*, and *GhPYE*, and this activation was independent of GhbHLH104 and GhbHLH115.

## 3. Discussion

### 3.1. Effects of Iron Deficiency Stress on Cotton Seedling Development

Soil salinization is a major environmental threat to the agriculture industry worldwide and affects approximately 20% of the world’s cultivated land and nearly half of all irrigated land, and the impact is becoming increasingly severe [59]. For example, the high pH in saline–alkali soil causes a lack of absorbable iron, which is harmful to the growth and development of seedling-stage cotton. Iron is required for a wide variety of plant metabolic processes but is particularly crucial for photosynthesis, as it acts as a cofactor in both photosystems [60,61]; consequently, Fe deficiency severely limits photosynthetic efficiency [62]. This study is the first to conduct a comprehensive and in-depth analysis of the photosynthetic efficiency of cotton seedlings under iron deficiency. Iron-deficient cotton seedlings exhibited leaf chlorosis due to reduced leaf Fe content and chlorophyll content, which in turn led to decreased leaf photosynthetic rate (Figure 1). There are two main mechanisms for the reduction in plant photosynthesis caused by abiotic stress: stomatal limitation and non-stomatal limitation [63,64]. Stomatal limitation refers to the decrease in stomatal conductance, which directly affects the photosynthesis [65]. Iron deficiency did not affect transpiration rate stomatal conductance and intercellular CO_2_ concentration (Appendix A). The non-stomatal limitations mainly include the destruction of chloroplast structure, the decrease in chlorophyll content, the inhibition of Ru-BP carboxylase activity, the decrease in glycolate oxidase activity, the decrease in photosynthetic phosphorylation activity, and the change in reactive oxygen species metabolism [62]. This reduction in photosynthetic rate in iron-deficient cotton leaves might be mainly attributable to the following two causes. First, with regard to the overall impact on transcriptional regulation, iron deficiency conditions inhibited the expression of genes involved in photosystem elements, including PSI, PSII, Cyt*b_6f_*, and Fd (Appendix A), with consequent weakening of the electron transfer efficiency of the photosynthetic chain. This pattern has previously been reported in *Arabidopsis* [38], phytoplankton [66,67], and sugar beet (*Beta vulgaris* L. cv. F58-554H1) [68]. Second, we observed the ultracellular structure of cotton leaf cells to be damaged under iron deficiency, predominantly on chloroplast and mitochondrial (Figure 4). Therefore, the decrease in photosynthetic rate under iron deficiency stress in cotton seedlings may be mainly caused by non-stomatal limiting factors. 

### 3.2. GhbHLH121 Is a Positive Regulator for the Response to Iron Deficiency in Cotton

The response of *AtbHLH121* to iron deficiency is a subject of debate (Figure 8) [45,46,47], but here we observed the expression of *GhbHLH121* to increase under iron deficiency (Figure 2). *AtbHLH121* was a transcription factor and positive regulator of iron homeostasis in *Arabidopsis* [45,46,47]. Under −Fe conditions, *Atbhlh121* mutants have decreased tolerance to iron deficiency, exhibiting strong inhibition of primary root growth and FCR activity along with low fresh weight and chlorophyll content [45,46,47].

We found that ectopic expression of *GhbHLH121* in *Arabidopsis* can increase the iron content of transgenic *Arabidopsis GhbHLH121*-*OE* and improve their tolerance to iron deficiency (Figure 5). Conversely, suppressing its expression in cotton resulted in the leaves becoming chlorotic, chlorophyll and leaf iron contents being reduced, and the photosynthetic rate decreasing (Figure 3). In addition, iron deficiency in cotton caused the ultracellular structure of leaves to be severely damaged, chloroplast thylakoids more loosely arranged, and mitochondria almost completely hollowed (Figure 4). These results suggest that inhibiting expression of *GhbHLH121* decreases cotton tolerance to iron deficiency.

In cotton, *GhbHLH121* is an iron deficiency response gene, which activates the ex-pression of downstream genes only when cotton faces iron deficiency. Guerinot et al. propose a model in which phosphorylated AtbHLH121 is allowed to accumulate when plants become Fe-deficient and interacts with subgroup IVc bHLH transcription factors. These heterodimers bind to the promoters of subgroup Ib genes and induce their expression. Subgroup Ib transcription factors then form heterodimers with FIT to activate the transcription of FRO2, IRT1, and other FIT-regulated genes. The induction of Fe uptake genes enhances Fe uptake and alleviates iron deficiency [45]. However, whether iron deficiency affects phosphorylation of GhbHLH121 in cotton remains to be seen.

### 3.3. The Transcriptional Regulatory Network of Cotton GhbHLH121

Due to whole-genome duplication and allopolyploidization, the cotton genome contains many more members of the bHLH transcription factor family than does *Arabidopsis* (Appendix A), and they also exhibit specific transcriptional patterns. In *Arabidopsis*, the bHLH Ib subgroup has four members whose expression is induced by iron deficiency [69]. In contrast, the cotton genome features seven members and only *GhbHLH38* is induced by iron deficiency (Table 1). Moreover, *GhbHLH101* is barely expressed in leaves (Appendix A). Among iron homeostasis genes, the core gene *FIT* (*bHLH29,* bHLH IIIe) has four homologs in the cotton genome, of which only a pair of duplicated homeologs are stably expressed in leaves (Appendix A). Of the bHLH IVc subgroup [70] in *Arabidopsis*, only *AtbHLH115* is induced by iron deficiency [71]. Meanwhile, cotton members of this subgroup comprise two homologs of *GhbHLH104* and ten homologs of *GhbHLH115* (Appendix A), with expression of *GhbHLH104* being significantly induced by iron deficiency in cotton seedlings (Table 1). There are conflicting reports regarding the transcriptional activity of *bHLH12* under iron deficiency in *Arabidopsis* [45,46,47]. Our study confirmed that *GhbHLH121* is induced under iron deficiency in cotton seedlings (Figure 2C). In general, bHLH transcription factors have been reported to play important roles in the regulation of plant iron homeostasis [50,51], and our results confirmed that cotton also uses members of this family to regulate iron homeostasis. However, the observed transcriptional differences indicate that cotton has formed an iron deficiency regulation network with its own distinct characteristics.

Few studies have reported on the regulation of response to iron deficiency in cotton [52]. In the present work, it was found that GhbHLH121, as an upstream iron deficiency-responsive protein, forms heterodimers with GhbHLH104, GhbHLH115, and GhPYE (Figure 5) that act to increase the expression of iron-responsive genes involved in regulation of cotton iron homeostasis (Table 1). The *35S_Pro_:GhbHLH121:GFP* signal was initially detected in both nucleus and cytoplasm (Figure 2E), but the presence of GhbHLH104, GhbHLH115, or GhPYE caused it to localize only to the nucleus (Figure 5C). GhbHLH121 does not have a transmembrane signal peptide, but does co-localize with GhGCN5 and GhNRT in the cytoplasm (Appendix A). The post-transcriptional regulation of GhbHLH121 remains to be further explored. All told, this and prior evidence indicate that bHLH121 is a positive iron regulator in both *Arabidopsis* [45,46,47] and cotton. 

While both cotton and *Arabidopsis* are dicotyledonous plants, we observed some notable differences in their respective iron regulatory networks (Figure 8, Appendix A). Under +Fe conditions, inhibiting the expression of *GhbHLH121* decreased expression of *GhbHLH38*, *GhFIT*, *GhPYE*, and *GhBTS*. Meanwhile, under −Fe conditions, the genes *GhbHLH38*, *GhFIT*, *GhPYE*, *GhBTS*, *GhIRT3*, *GhFRO2*, *GhAHA2*, *GhNAS4*, and *GhYSL1* all exhibited significantly lower expression than in controls, but no significant difference was observed in the expression of *GhbHLH101* (Table 1). This suggests that in cotton, *GhbHLH101* is not involved in the regulation of iron homeostasis by GhbHLH121, which is different from prior reports in *Arabidopsis* [45,46,47]. We also found GhbHLH121 to activate transcription of *GhFIT*, *GhbHLH38*, and *GhPYE* (Figure 7A–E), which is not consistently attested in *Arabidopsis* [45,46,47]. Gao et al. and Kim et al. found that bHLH121 did not bind the promoter of *FIT*, but Lei et al. reported bHLH121 to bind the *FIT* promoter and interact with bHLH IVc TFs, which may activate the binding on the *FIT* promoter [45,46,47]. However, bHLH121 alone had no notable effect on *FIT* and *bHLH38* promoter activity, while bHLH104 and bHLH115 each induced significant activation [46]. In addition, bHLH IVc TFs were reported to directly bind the *bHLH121* promoter [46], which is different to our results. Here we found that GhbHLH121 directly binds to the promoters of *GhbHLH38*, *GhFIT*, and *GhPYE* and activates their transcription independent of GhbHLH104 and GhbHLH115 (Figure 7A–E). *GhbHLH104* and *GhbHLH115* were both induced by iron deficiency, but their expression was not associated with *GhbHLH121* (Figure 7F–H and Figure 8).

Ultimately, our work revealed that GhbHLH121 has a positive regulatory role in the iron homeostasis network (Figure 8). As an upstream iron deficiency-responsive protein, GhbHLH121 forms a heterodimer with GhbHLH104, GhbHLH115, and GhPYE. It also activates the expression of *GhbHLH38*, *GhFIT*, and *GhPYE* independent of GhbHLH104 and GhbHLH115 and regulates the expression of Fe uptake genes (such as *GhIRT3*, *GhAHA2,* and *GhFRO2*) and Fe transport genes (such as *GhYSL1* and *GhNAS4*).

## 4. Materials and Methods 

### 4.1. Plant Materials and Growth Conditions 

Cotton plants (*Gossypium hirsutum*, accession Texas Marker-1 [TM-1]) were grown in a growth chamber with a 16/8 h light/dark cycle at 28 °C/25 °C and 70% relative humidity. Cotton seeds were sterilized and germinated by the paper towel method. In brief, cotton seeds were sterilized with 3% H_2_O_2_ for 30 min, rinsed 4–5 times with ddH_2_O, and then laid flat on soaked germinating paper. The germinating paper was rolled into a tube and placed in a germinating bag, which was then placed vertically in an incubator for four days. Afterwards, the seedlings were transferred to a hydroponic bucket filled with water for three days and then incubated with ^1^/_2_ Hoagland nutrient solution with 75 µM Fe^2+^-EDTA (+Fe) or without it (−Fe) for two weeks. 

*Arabidopsis thaliana* ecotype Columbia (Col-0) seedlings were grown in a growth chamber with a 16/8 h light/dark cycle at 22 °C and 70% relative humidity. Briefly, seeds were sterilized and transferred to plates with half-strength Murashige and Skoog (MS) medium. After stratification at 4 °C for two days, the plates were transferred to the growth chamber. Fe-sufficient medium (+Fe) consisted of ^1^/_2_ MS medium with 1% (*w*/*v*) sucrose, 0.8% (*w*/*v*) agar, and 75 µM Fe^2+^-EDTA, while Fe-deficient medium (−Fe) consisted of the same ingredients without Fe^2+^-EDTA. 

### 4.2. Construction of Materials for Virus-Induced Gene Silencing (VIGS-GhbHLH121) in TM-1

VIGS was performed following a previously described protocol [72]. The 1–300 bp nucleotide sequence of *GhbHLH121* was amplified by specific primers (Appendix A) from TM-1 cDNA and cloned into a *pTRV2* vector digested with *Eco*RI and *Bam*HI. The resulting construct was transformed into *Agrobacterium tumefaciens* GV3101 to obtain the *TRV2:GhbHLH121* bacterial solution. Cotton seeds were sterilized and germinated by the paper towel method, transferred to a hydroponic bucket filled with water for three days, and finally inoculated with *TRV2:GhbHLH121*, *TRV2:CLA* (positive control), or *TRV2:00* (negative control) bacterial solution.

### 4.3. Generation of Transgenic Arabidopsis

The full-length coding sequence of *GhbHLH121* was amplified by specific primers (Appendix A) from TM-1 cDNA and fused with *GUS* to generate *35S:GhbHLH121:GUS* vectors (*GhbHLH121-OE*). The promoter of *GhbHLH121* was amplified by specific primers (Appendix A) from TM-1 DNA and inserted into the *pBI121* vector digested with *HindIII* and *XbaI* to generate GhbHLH121pro:GUS vectors. The constructs were then transformed into *Agrobacterium tumefaciens* strain GV3101 and injected into Col-0 *Arabidopsis* by the floral dip method. Positive transgenic plants were selected on ^1^/_2_ MS medium containing 50 mg/mL kanamycin. T3 plants were used for phenotype analysis. Primers used for the vector construction are listed in Appendix A.

### 4.4. Real-Time Quantitative Reverse Transcription PCR (qRT-PCR)

Total RNA was extracted by means of the Biospin Plant Total RNA Extraction Kit (BioFlux, Cluj, Romania). The RNA was then used for the synthesis of first-strand cDNA with the SuperScript III reverse transcriptase kit (Invitrogen, Waltham, MA, USA). Afterwards, qRT-PCR was performed using the Light Cycler FastStart DNA Master SYBR Green I Kit (Roche, Basel, Switzerland) according to the manufacturer’s instructions (Bio-Rad, Hercules, CA, USA). Means and standard deviations were calculated from three biological replicates. Primers used for the qRT-PCR are listed in Appendix A. 

### 4.5. Phylogenetic Analysis

For phylogenetic analysis, protein sequences were retrieved from TAIR and COTTONOMICS. Sequences of bHLH transcription factors involved in Fe homeostasis were aligned using ClustalX 1.5. The phylogenetic tree included multiple plant species and was constructed using MEGA 5 with the NJ method, and the selected model was the Poisson model with 1000 bootstrap replications [73]. We used the ITOL9 web server to beautify the resulting phylogenetic tree.

### 4.6. Measurement of Iron Deficiency-Associated Physiological Parameters

#### 4.6.1. Photosynthetic Measurements

After growing cotton seedlings in +Fe or −Fe solution, the penultimate leaf was placed in the 3 cm leaf chamber of an LI-6400 portable photosynthesizer (LI-6400XT; Li-Cor, Darmstadt, Germany) for evaluation of photosynthetic parameters. Photosynthesis rate, stomatal conductance, transpiration rate, and intercellular CO_2_ concentration were determined simultaneously. Conditions in the measurement chamber consisted of a flow rate of 400 mol/s, CO_2_ concentration in the sample chamber of 400 mmol/mol, and relative humidity of 60%. Means and standard deviations were calculated from three biological replicates.

#### 4.6.2. Fe Concentration Measurement

To determine Fe concentration, samples (*Arabidopsis* seedlings or cotton leaves) were harvested and dried at 65 °C for three days. About 100 mg dry weight was mixed with 5 mL of 1 M HNO_3_ and 1 mL of H_2_O_2_ for 20 min at 220 °C. Each sample was then diluted to 16 mL with ultrapure water and the Fe concentration determined using a Thermo SCIENTIFIC ICP-MS (iCAP6300, Waltham, MA, USA). Means and standard deviations were calculated from three biological replicates.

#### 4.6.3. Ferric Chelate Reductase Assays

To determine ferric chelate reductase activity (FCR), cotton leaves were harvested and placed in assay solution containing 0.1 mM Fe^3+^-EDTA and 0.3 mM ferrozine in the dark for 20 min. An identical assay solution without the addition of plant tissue was used as a blank. After incubation, the absorbance of the purple-colored Fe(II)−ferrozine complex was measured at 562 nm and concentration calculated using a molar extinction coefficient of 28.6 mM^−1^ cm^−1^ [74]. Means and standard deviations were determined from three biological replicates.

#### 4.6.4. Chlorophyll Measurement

To determine chlorophyll content, 0.1 g samples of cotton penultimate leaves were harvested and placed in 80% acetone solution for 30 min. The absorbance (A) was measured at 663 and 646 nm. Total chlorophyll content (expressed as μg/g FW) was then calculated using the following equations: Chl a = 12.25 × A663 − 2.79 × A646, Chl b = 21.50 × A646 − 5.10 × A663, and Chl a + Chl b = 7.05 × A663 + 18.09 × A646 [75]. Means and standard deviations were determined from three biological replicates.

#### 4.6.5. NBT Staining and DAB Staining 

Cotton leaves were put into a tube with NET staining solution (G1023, 100 mL, Solarbio, Beijing, China) or DAB staining solution (G1022, 100 mL, Solarbio, Beijing, China), with the leaves being completely immersed in the solution. Leaves were subjected to 15 min of vacuumization and then incubated overnight at room temperature. The next day, the leaves were decolorized with absolute ethanol for 10 min, 95% for 10 min, and then 70% until completely decolorized.

#### 4.6.6. Measurements of SOD, POD, and CAT Activity 

To determine physiological parameters of oxidative defense, 0.1 g cotton penultimate leaf was used in the SOD Quantification Assay Kit (BC0170, Solarbio, Beijing, China), POD Quantification Assay Kit (BC0095, Solarbio, Beijing, China), and CAT Quantification Assay Kit (BC0205, Solarbio, Beijing, China) as described by the manufacturer. Means and standard deviations were calculated from three biological replicates.

### 4.7. Microscopic Observation

#### 4.7.1. Subcellular Localization

To produce the constructs *35S_Pro_:GhbHLH121:GFP*, *35S_Pro_:GhNRT:GFP*, and *35S_Pro_:GhGCN5:GFP*, full-length coding sequences of *GhbHLH121, GhNRT*, and *GhGCN5* were inserted into the *pBINGFP4* vector. The constructs were transformed into *Agrobacterium tumefaciens* strain GV3101 and subsequently injected into three-week-old *Nicotiana benthamiana* leaves. After cultivation in the dark for two days, fluorescence in the epidermal cells was visualized on a confocal microscope (Zeiss LSM780, Oberkochen, Germany) with the excitation laser at 488 nm. The primers used for vector construction are listed in Appendix A. For each combination, three biological replicates were performed.

#### 4.7.2. Transmission Electron Microscopy

For TEM analysis of treated plants, cotton penultimate leaves were immersed in a solution containing 2.5% glutaraldehyde. The leaves were fixed with osmic acid for two hours, rinsed with phosphate buffer solution, and then subjected to ethanol gradient dehydration. After embedding with an embedding agent overnight, thin sections were prepared with an ultrathin microtome, stained with uranyl acetate and lead citrate, and examined using a Hitachi H-7650 transmission electron microscope at 80 kV.

### 4.8. Validation of Protein–Protein Interactions

#### 4.8.1. Yeast Two-Hybrid Screening and Assays 

For the yeast two-hybrid screening, the full-length coding sequence of *GhbHLH121* was inserted into *pGBKT7* as the bait vector and transformed into yeast strain Y2H to produce the bait strain. Cotton leaf and root cDNA libraries cloned into the *pGADT7-Rec* vector were used to produce the bait strain with the Make Your Own “Mate & Plate” Library System (Cat. No. 630490, TaKaRa, Dalian, China). The library strain was then mated with the bait strain, and the mating mixture was spread onto SD/-Trp/-Leu/-His medium and incubated at 30 °C for 8–10 days. Positive clones were amplified by Matchmaker Insert Check PCR Mix 2 (Cat. No. 630497, TaKaRa, Dalian, China).

For the yeast two-hybrid assay, the full-length coding sequence of *GhbHLH121* was inserted into *pGBKT7* as the bait vector, and the full-length coding sequences of *GhbHLH104, GhbHLH115,* and *GhPYE* were inserted into *pGADT7* to produce the prey vectors. The bait vector and each prey vector were introduced together into Y2HGlod, and the transformed cells were spread onto SD/-Trp/-Leu/-His medium and incubated at 30°C for 2–3 days. The combination of *pGBKT7-53* and *pGADT7-T* was used as a positive control and the combination of *pGBKT7-Lam* and *pGADT7-T* as a negative control (Matchmaker Gold Yeast Two-Hybrid System, Cat. No. 630489, TaKaRa, Dalian, China). Primers used for the vector construction are listed in Appendix A. For each combination, three biological replicates were performed.

#### 4.8.2. Bimolecular Fluorescence Complementation 

The full-length coding sequence of *GhbHLH121* was inserted adjacent to the sequence for the N-terminal fragment of YFP, denoted as *YNE-GhbHLH121*. The full-length coding sequences of *GhbHLH104, GhbHLH115*, and *GhPYE* were similarly inserted alongside the sequence for the C-terminal fragment of YFP, and the resulting constructs were denoted *35S_Pro_:GhbHLH104:YCE*, *35S_Pro_:GhbHLH115:YCE*, *35S_Pro_:GhPYE:YCE,* respectively. The constructs were transformed into *A. tumefaciens* strain GV3101 and bacteria harboring the effectors and reporters were mixed in a 1:1 volume ratio. The obtained mixtures were then injected into the abaxial sides of three-week-old *N. benthamiana* leaves. After two days of cultivation in the dark, epidermal cells were imaged on a confocal microscope (Zeiss LSM780, Oberkochen, Germany), with the YFP signal elicited by a 514 nm excitation laser and recorded. Primers used for the vector construction are listed in Appendix A. For each combination, three biological replicates were evaluated.

#### 4.8.3. Luciferase Complementation Imaging Assay 

The full-length coding sequence of *GhbHLH121* was inserted alongside the sequence encoding the C-terminal fragment of luciferase, producing the construct *cLUC-GhbHLH121*. The full-length coding sequences of *GhbHLH104, GhbHLH115*, and *GhPYE* were likewise inserted alongside that encoding the N-terminal fragment of luciferase, which constructs were denoted as *GhbHLH104-nLUC*, *GhbHLH115-nLUC*, and *GhPYE-nLUC,* respectively. The constructs were transformed into *A. tumefaciens* strain GV3101 and bacteria harboring the effectors and reporters were mixed in a 1:1 volume ratio. The resulting mixtures were then injected into the abaxial sides of three-week-old *N. benthamiana* leaves. After two days of cultivation in the dark, Luciferin (100 μM) spray (Sigma, 103404-75-7, St. Louis, MO, USA) was smeared on the backs of the leaves and luciferase signals were obtained by a low-light cooled charge-coupled device imaging apparatus (Tanon 5200, Shanghai, China). The positive controls were FTL9 and FD1 [76]. Primers used for the vector construction are listed in Appendix A. For each combination, three biological replicates were assessed.

### 4.9. Validation of DNA–Protein Interactions

#### 4.9.1. Yeast One-Hybrid Assay 

E-box-containing fragments of the promoters for *GhbHLH38, GhFIT,* and *GhPYE* were inserted into vectors to produce bait vectors and an mE-box containing a mutation of CA to TC [58], which were then digested by *BstB*I and the linearized plasmids transferred into yeast strain Y1H gold to yield the *pBait-AbAi* bait strain. Yeasts were screened according to the concentration of AbAi. The full-length coding sequence of *GhbHLH121* was inserted into the *pGBKT7* vector to construct the prey vector, which was then introduced into the *pBait-pAbAi* strain and the bacteria spread on SD/-Leu/AbA and incubated at 30°C for two to three days. The yeast one-hybrid assay was then conducted using the Matchmaker Gold Yeast One-Hybrid Library Screening System kit (Cat. No. 630491, TaKaRa, Dalian, China). For each combination, three biological replicates were performed.

#### 4.9.2. Luciferase Assay 

The promoters of *GhbHLH38, GhFIT,* and *GhPYE* were inserted into the *pGreen II 0800-LUC* vector to create reporter constructs. The full-length coding sequence of *GhbHLH121* was inserted into the *pGreen II 62SK* vector as the effector. The constructs were transformed into *A. tumefaciens* strain GV3101 and bacteria harboring the effectors and reporters were mixed in a 1:1 volume ratio. The resulting mixtures were then injected into the abaxial sides of three-week-old *N. benthamiana* leaves. After two days of cultivation in the dark, luciferase signals were obtained by observing leaves coated with Luciferin (100 μM) spray (Sigma, 103404-75-7, St. Louis, MO, USA) under a low-light cooled charge-coupled device imaging apparatus (Tanon 5200, Shanghai, China). The LUC/REN ratio was then determined using a dual-luciferase reporter gene analysis system (E2490, Promega, Madison, WI, USA). For each combination, three biological replicates were evaluated. Primers used for the vector construction are listed in Appendix A. Means and standard deviations were calculated from three biological replicates.

## Figures and Tables

**Figure 1 plants-12-01955-f001:**
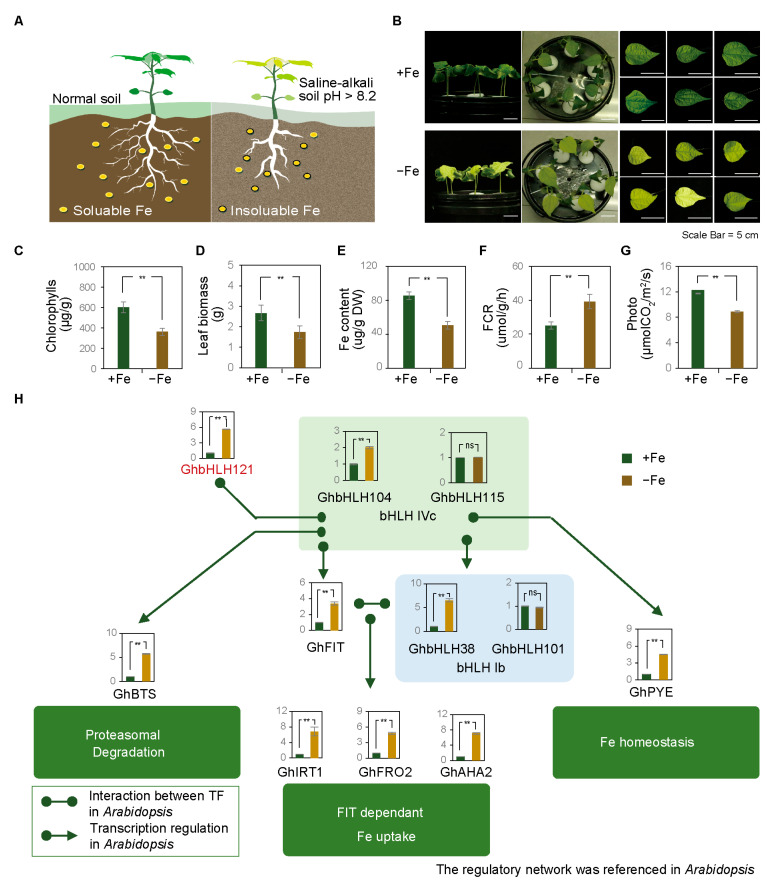
Iron deficiency induces differential expression of cotton iron deficiency regulatory genes, resulting in iron deficiency phenotypes. (**A**) Schematic diagram of cotton seedling growing on normal (left) and saline–alkali (pH > 8.2, right) soil. Iron exists in form of soluble iron under normal soil conditions, which can be absorbed by plants, while in saline–alkali soil, iron mainly exists as insoluble iron, which leads to cotton suffering from iron deficiency stress, noted as chlorosis of cotton leaves. (**B**) Representative images of the phenotypes of cotton seedlings grown for two weeks under +Fe (^1^/_2_ Hoagland nutrient solution) or −Fe (^1^/_2_ Hoagland nutrient solution without Fe) conditions. Statistics were determined from three biological replicates, and each experiment contained ten seedlings. Scale bar = 5 cm. (**C**–**G**) Histograms of chlorophyll content (**C**), leaf biomass (**D**), Fe content (**E**), ferric chelate reductase activity (FCR) (**F**), and net photosynthesis rate (G) in cotton seedling leaves. Seedlings were grown for two weeks in +Fe or −Fe conditions. (**H**) Expression of marker genes of the cotton iron deficiency regulatory genes. Relative expression was determined by qRT-PCR of genes in cotton seedling leaves. Seedlings were the same plants as in (**C**–**G**). The regulatory relationship between genes refers to the regulatory network of *Arabidopsis* and does not mean its presence in cotton. Values represent means ± SD of three biological replicates. Significant differences were determined by Student’s *t*-test, ** *p* < 0.01.

**Figure 2 plants-12-01955-f002:**
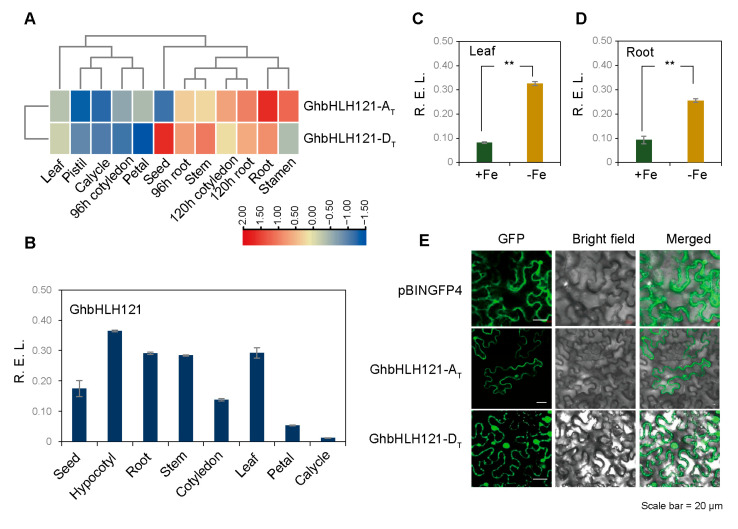
*GhbHLH121* encodes a transcription factor involved in the regulation of response of iron deficiency that is expressed in cotton root and shoot tissues. (**A**) Heat map of *GhbHLH121* expression in different cotton tissues. The tissues used for expression profiling are indicate at the bottom and the genes on the left. The heatmap was constructed using TBtools. The color of each box represents the transcription level of the gene. The values are presented using the relative expression change after row normalization. (**B**) Histogram of *GhbHLH121* expression as determined by qPCR in different cotton seedling tissues. (**C**,**D**) Histogram of *GhbHLH121* expression as determined by qPCR in the leaf (**C**) and root (**D**) of cotton seedlings grown for two weeks in +Fe or −Fe solution. (**E**) Subcellular localization of GhbHLH121-A_T_ and GhbHLH121-D_T_. Each experiment utilized three biological replicates. Scale bar = 20 µm. Values represent means ± SD of three biological replicates. Significant differences were determined by Student’s *t*-test, *** p* < 0.01.

**Figure 3 plants-12-01955-f003:**
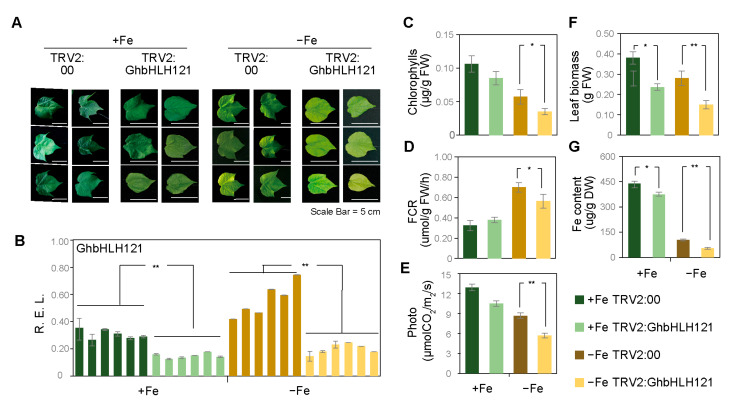
*TRV2:GhbHLH121* seedlings exhibit reduced tolerance to iron deficiency. (**A**) Representative images showing the phenotypes of *TRV2:00* and the *TRV2:GhbHLH121* seedlings grown for three weeks in +Fe or −Fe solution. (**A**) The last leaves of seedlings from *TRV2:00* and the *TRV2:GhbHLH121* seedlings. Statistics were determined from three biological replicates, and each experiment contained ten seedlings. Scale bar = 5 cm. (**B**) Expression of *GhbHLH121* in *TRV2:00* and the *TRV2:GhbHLH121* seedlings in +Fe or −Fe solution. Expression was determined by qPCR using RNA from leaf of (**A**). (**C**–**G**) Histograms of chlorophyll content (**C**), FCR activity (**D**), net photosynthesis rate (**E**), leaf biomass (**F**), and Fe content (**G**) in seedlings from (**A**). Photo: net photosynthesis rate. Relative expression was determined by qRT-PCR in cotton leaves. Values represent means ± SD of three biological replicates. Significant differences were determined by Student’s *t*-test, * *p* < 0.05, ** *p* < 0.01.

**Figure 4 plants-12-01955-f004:**
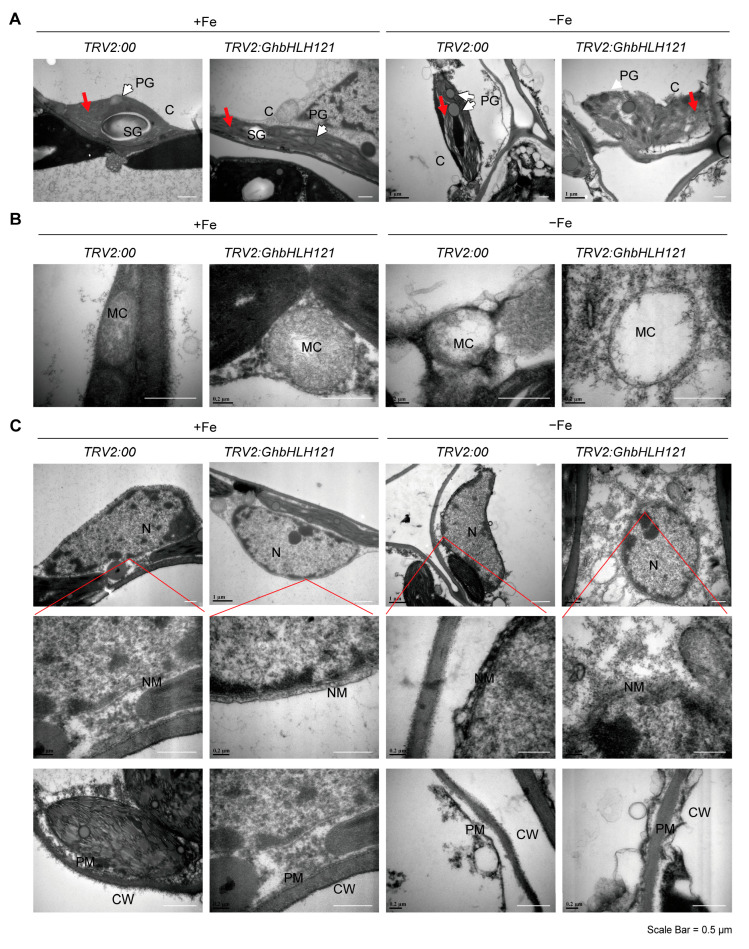
Suppression of *GhbHLH121* increases cell ultrastructure damage due to iron deficiency in the leaves of cotton seedlings. Representative images of cell ultrastructures such as chloroplast (**A**), mitochondrial (**B**), and nucleus (**C**) examined with TEM in leaf cell cross-sections from the VIGS cotton seedlings assay of Figure 3. C: chloroplasts; SG: starch grains; PG: plastoglobules; MC: mitochondria; CW: cell wall; PM: plasma membrane; NM: cell nuclear membrane. Red arrows = stacked thylakoids in grana. White arrows = plastoglobules. Red line = the position of the enlarged cell membrane on the nucleus. Scale bar = 0.5 µm.

**Figure 5 plants-12-01955-f005:**
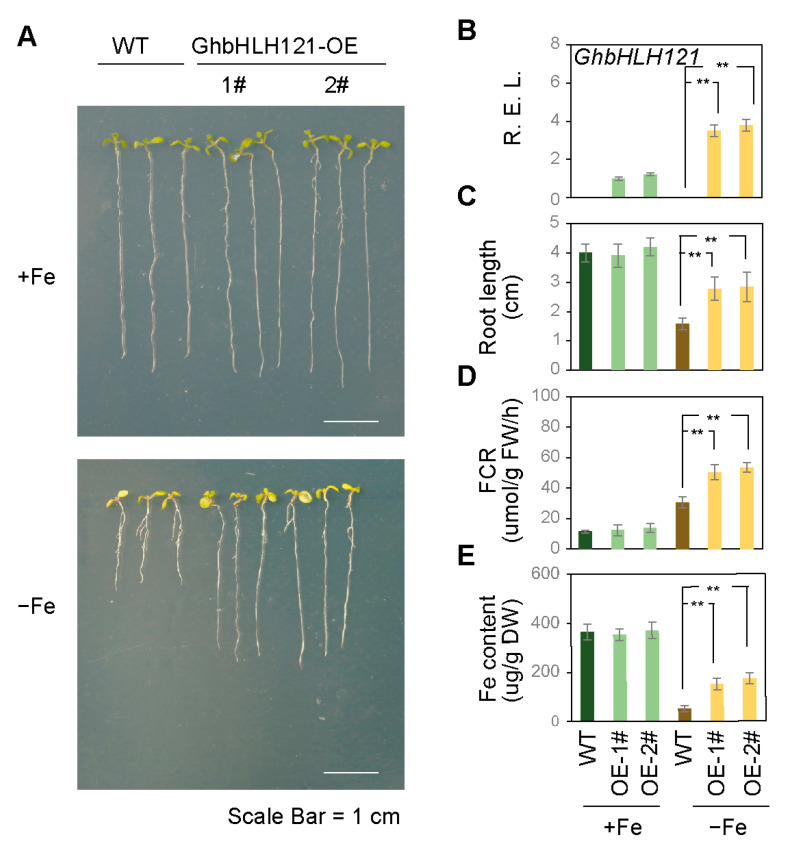
Ectopic expression of *GhbHLH121* in *Arabidopsis* seedlings conferred enhanced tolerance to iron deficiency. (**A**) Representative images of *Arabidopsis thaliana* wild type (WT) and *GhbHLH121-OE* seedlings grown for 7 days under +Fe (^1^/_2_ MS with Fe^2+^) or −Fe (^1^/_2_ MS without Fe^2+^) conditions. Statistics were determined from three biological replicates, and each experiment contained 15 seedlings. Scale bar = 1 cm. (**B**–**E**) Histogram of *GhbHLH121* expression (**B**), root length (**C**), FCR (**D**), and Fe content (**E**) in seedlings from (**A**). Relative expression was determined by qRT-PCR. Values represent means ± SD of three biological replicates. Significant differences were determined by Student’s *t*-test, ** *p* < 0.01.

**Figure 6 plants-12-01955-f006:**
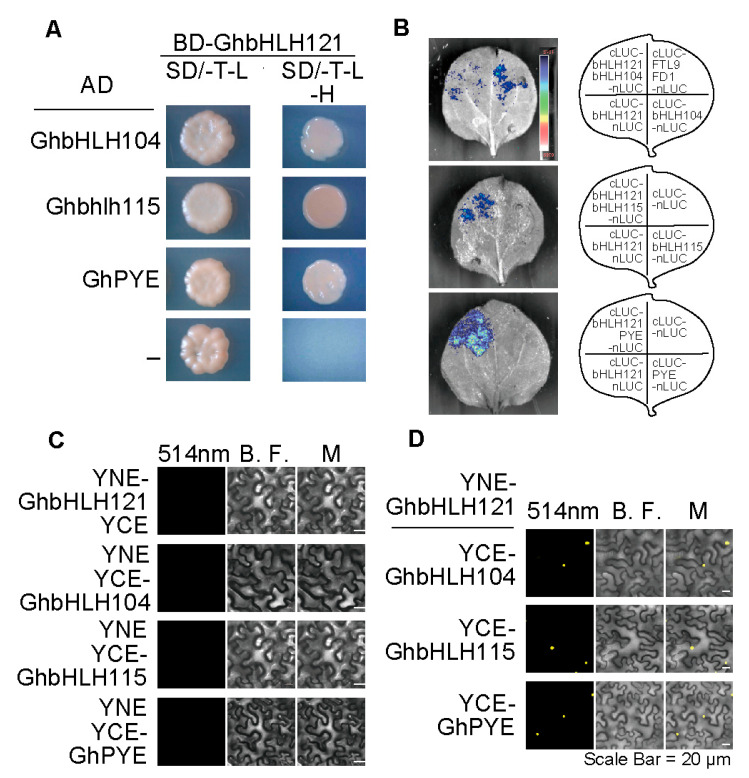
GhbHLH121 can interact with GhbHLH IVc family proteins and GhPYE. (**A**) Yeast two-hybrid assays. *GhbHLH104*, *GhbHLH115,* and *GhPYE* were fused with the GAL4 activation domain (AD) as prey, and *GhbHLH121* with the GAL4 DNA binding domain (BD) as bait. Interaction was indicated by the ability of cells to grow on synthetic dropout (SD) medium lacking T/L/H. T, tryptophan; L, leucine; H, histidine. (**B**) Luciferase Complementation Imaging Assay (LCI) assays. *GhbHLH104*, *GhbHLH115,* and *GhPYE* were fused with the N-terminus of LUC (LUC-N) and *GhbHLH121* with the C-terminus (LUC-C), then transferred into *N. benthamiana* leaves. The positive controls were FTL9 and FD1. (**C**,**D**) Bimolecular fluorescence complementation (BiFC) assays. *GhbHLH104*, *GhbHLH115,* and *GhPYE* were fused with the N-terminus of YFP (YNE) and *GhbHLH121* with the C-terminus of YFP (YCE), then transferred into *N. benthamiana* leaves and visualized by confocal microscopy. B. F.: bright field. M: merge. Scale bar = 20 µm.

**Figure 7 plants-12-01955-f007:**
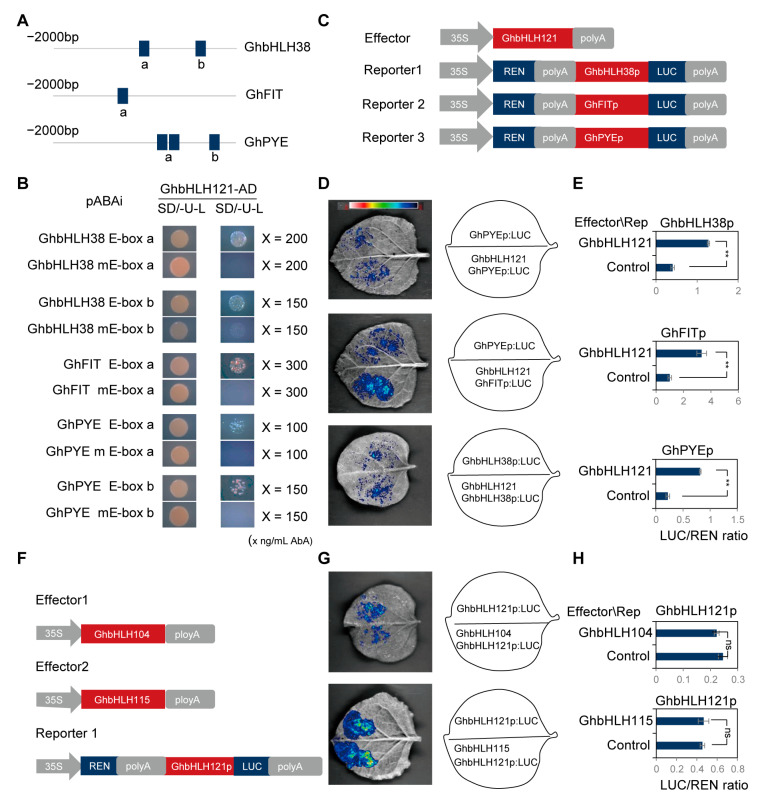
GhbHLH121 directly activates the transcription of *GhbHLH38*, *GhFIT*, and *GhPYE* independent of GhbHLH104 and GhbHLH115. (**A**) Schematic diagrams showing the promoter structures of *GhbHLH38, GhFIT*, and *GhPYE*. Blue boxes indicate E-box motifs. (**B**) Image of yeast one-hybrid assays. Each experiment utilized three biological replicates. (**C**) Schematic diagram of the constructs used in the transient expression assay in (**D**). *GhbHLH38, GhFIT*, and *GhPYE* promoters were fused with *pGreen 0800LUC* as reporters, and *GhbHLH121* with *pGreen II 62SK* as the effector. (**D**) Photo of luciferase imaging in *N. benthamiana* leaves with different combinations of the effector and reporters. Each experiment utilized three biological replicates. (**E**) Histogram of relative reporter activity (LUC/REN) in *N. benthamiana* leaves expressing the indicated reporters and effectors, detailed in (**D**). The LUC/REN ratio represents LUC activity relative to that of the internal control (REN driven by the 35S promoter). (**F**) Schematic diagram of the constructs used in the transient expression assay in (**G**). The *GhbHLH121* promoter was fused with *pGreen0800LUC* as the reporter, and *GhbHLH104* and *GhbHLH115* with *pGreen II 62SK* as effectors. (**G**) Photo of luciferase imaging in *N. benthamiana* leaves with different combinations of effectors and the reporter. Each experiment utilized three biological replicates. (**H**) Histogram of relative reporter activity (LUC/REN) in *N. benthamiana* leaves expressing the indicated reporters and effectors, detailed in (**G**). Values represent means ± SD of three biological replicates. Significant differences were determined by Student’s *t*-test, ** *p* < 0.01, ns = no significance.

**Figure 8 plants-12-01955-f008:**
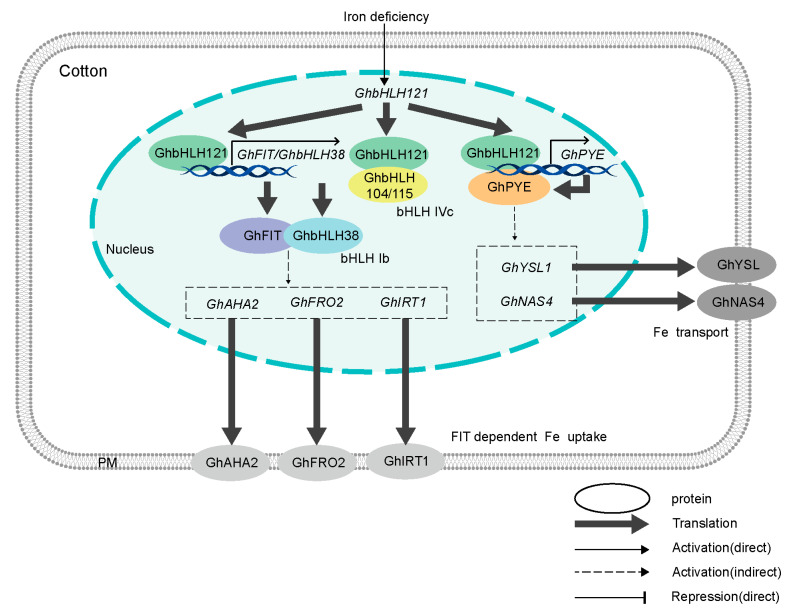
A working model illustrating the roles of GhbHLH121 in response to iron deficiency in cotton. When cotton is subjected to iron deficiency, expression of *GhbHLH121* is increased, though GhbHLH104 and GhbHLH115 do not activate its transcription. GhbHLH121 forms a heterodimer with GhbHLH104 or GhbHLH115, but can activate the expression of *GhbHLH38*, *GhFIT*, and *GhPYE* independent of them.

**Table 1 plants-12-01955-t001:** Differential expression of Fe deficiency-responsive genes in *TRV2:00* and the *TRV2:GhbHLH121* seedlings in +Fe or −Fe solution. The relative gene expression levels measured by qPCR in leaves of the *TRV2:00* and the *TRV2:GhbHLH121* seedlings in +Fe or −Fe solution. The expression of GhHistone3 was used to normalize mRNA levels, and the gene expression level in the *TRV2:00* under Fe-sufficient conditions was set to 1. Values represent means ± SD of three biological replicates. Significant differences from the corresponding wild type are indicated by an asterisk (* *p* < 0.05, ** *p* < 0.01), as determined by Student’s *t*-test.

Genes	TRV2:GhbHLH121 +Fe	TRV2:00 -Fe	TRV2:GhbHLH121 -Fe
Subgroup bHLH IVc genes			
*GhbHLH104-A10*	1.07 ± 0.24	2.46 ± 0.34	2.54 ± 0.32
*GhbHLH104-D10*	0.84 ± 0.19	2.29 ± 0.51	2.18 ± 0.30
*GhbHLH115 A01*	1.01 ± 0.21	2.36 ± 0.33	2.34 ± 0.31
*GhbHLH115 D01*	0.95 ± 0.15	1.25 ± 0.12	1.12 ± 0.16
*GhbHLH115 A01-2*	0.82 ± 0.11	0.78 ± 0.11	0.83 ± 0.10
*GhbHLH115 D01-2*	0.97 ± 0.11	0.90 ± 0.13	0.85 ± 0.12
*GhbHLH115 A05*	1.18 ± 0.17	1.28 ± 0.15	1.33 ± 0.17
*GhbHLH115 D05*	0.93 ± 0.14	1.10 ± 0.12	1.06 ± 0.14
*GhbHLH115 A11*	0.87 ± 0.12	0.84 ± 0.11	0.94 ± 0.11
*GhbHLH115 D11*	0.99 ± 0.13	0.94 ± 0.13	0.98 ± 0.12
*GhbHLH115 A13*	1.14 ± 0.12	1.01 ± 0.15	0.94 ± 0.13
*GhbHLH115 D13*	0.93 ± 0.12	0.97 ± 0.12	0.95 ± 0.13
Subgroup bHLH Ib genes			
*GhbHLH101 A11*	0.94 ± 0.12	1.01 ± 0.12	0.92 ± 0.13
*GhbHLH101 D11*	0.91 ± 0.13	1.07 ± 0.12	1.01 ± 0.14
*GhbHLH101 D11-2*	0.90 ± 0.16	1.15 ± 0.12	1.23 ± 0.15
*GhbHLH38 A05*	0.89 ± 0.23	58.80 ± 9.11	33.86 ± 7.88 **
*GhbHLH38 D05*	0.80 ± 0.07 **	133.96 ± 17.10	53.94 ± 11.95 **
*GhbHLH38 A09*	0.85 ± 0.20	57.27 ± 6.11	32.40 ± 5.67 **
*GhbHLH38 D09*	0.92 ± 0.14	72.90 ± 8.12	65.44 ± 5.76 *
Fe uptake genes			
*GhFIT A05*	0.48 ± 0.03 **	5.61 ± 0.06	0.87 ± 0.15 **
*GhFIT D05*	0.31 ± 0.01 **	15.61 ± 0.14	7.58 ± 1.09 **
*GhFIT A09*	0.63 ± 0.03 **	1.75 ± 0.18	1.03 ± 0.13 **
*GhFIT D09*	0.71 ± 0.10 **	1.97 ± 0.15	1.11 ± 0.20 *
*GhFRO2*	0.95 ± 0.14	8.68 ± 1.22	4.08 ± 0.56 **
*GhIRT3*	1.07 ± 0.01	5.60 ± 1.14	2.34 ± 0.35 **
*GhAHA2*	1.03 ± 0.04	4.69 ± 1.13	3.06 ± 0.62 **
Fe translocation genes			
*GhPYE*	1.05 ± 0.02	102.42 ± 12.07	73.34 ± 13.72 *
*GhYSL1*	1.14 ± 0.09	0.42 ± 0.11	0.31 ± 0.05 *
*GhNAS4*	0.88 ± 0.17	15.80 ± 0.21	5.00 ± 1.11 **
*GhAtZIF1*	0.93 ± 0.08	4.15 ± 0.17	4.34 ± 0.55
*GhFRD3*	1.00 ± 0.29	2.14 ± 0.43	2.22 ± 0.28
Fe homeostasis regulation genes			
*GhBTS A04*	0.55 ± 0.12 **	2.62 ± 0.27	0.91 ± 0.15 **
*GhBTS A11*	0.39 ± 0.04 **	1.81 ± 0.35	1.09 ± 0.14 **
*GhMYB72 A01*	0.89 ± 0.13	3.52 ± 0.42	3.24 ± 0.47
*GhMYB72 A08*	0.99 ± 0.18	7.34 ± 0.33	5.84 ± 0.98 *

**Table 2 plants-12-01955-t002:** Differential expression of Fe deficiency-responsive genes in WT and *GhbHLH121-OE* seedlings in +Fe or −Fe solution. The relative gene expression levels measured by qPCR in WT and *GhbHLH121-OE* seedlings grown for 7 days under +Fe or −Fe conditions. The expression of TUB2 was used to normalize mRNA levels, and the gene expression level in the wild type under Fe-sufficient conditions was set to 1. Values represent means ± SD of three biological replicates. Significant differences from the corresponding wild type are indicated by an asterisk (* *p* < 0.05, ** *p* < 0.01), as determined by Student’s *t*-test.

Genes	GhbHLH121OE +Fe	WT -Fe	GhbHLH121OE -Fe
Subgroup bHLH IVc genes			
*AtbHLH34*	1.34 ± 0.13	1.23 ± 0.17	1.53 ± 0.15
*AtbHLH104*	0.87 ± 0.12	0.89 ± 0.11	1.01 ± 0.11
*AtbHLH105*	1.02 ± 0.06	0.88 ± 0.12	1.29 ± 0.11
*AtbHLH115*	1.08 ± 0.15	2.03 ± 0.13	2.01 ± 0.25
Subgroup bHLH Ib genes			
*AtbHLH38*	4.46 ± 0.76 **	490.66 ± 36.56	1980.02 ± 62.31 **
*AtbHLH39*	0.80 ± 0.13	15.13 ± 3.10	40.96 ± 3.92 **
*AtbHLH100*	4.40 ± 0.52 **	295.60 ± 34.55	2234.81 ± 37.54 **
*AtbHLH101*	2.81 ± 0.32 **	19.59 ± 2.35	57.68 ± 8.48 **
Fe uptake genes			
*AtFIT*	2.01 ± 0.37 **	5.62 ± 0.25	10.84 ± 0.71 **
*AtFRO2*	2.70 ± 0.05 **	27.32 ± 0.34	39.78 ± 3.47 **
*AtIRTl*	4.44 ± 0.30 **	133.25 ± 11.56	230.75 ± 16.92 **
*AtIAHA2*	1.03 ± 0.17	2.68 ± 0.13	6.08 ± 0.34 **
Fe translocation genes			
*AtPYE*	1.64 ± 0.23	3.02 ± 0.20	9.69 ± 0.38 **
*AtYSL1*	0.96 ± 0.04	0.37 ± 0.12	0.37 ± 0.04
*AtNAS4*	2.34 ± 0.04 **	7.72 ± 0.29	14.56 ± 0.98 **
*AtZIF1*	1.53 ± 0.17 *	1.82 ± 0.19	2.92 ± 0.23 **
*AtFRD3*	0.81 ± 0.13	0.62 ± 0.10	1.05 ± 0.07**
Fe homeostasis regulation genes			
*AtBTS*	1.72 ± 0.15 **	1.54 ± 0.21	5.96 ± 0.19 **
*AtMYB10*	0.98 ± 0.19	6.73 ± 0.12	36.19 ± 0.85 **

## Data Availability

Accession numbers: Sequence data for the genes described in this study can be found in the GenBank database under the following accession numbers: XP_016714640.1(*GhbHLH121*), XP_016753020.1 (*GhbHLH38*), XP_016699442.2 (*GhbHLH104*), XP_016702823.1 (*GhbHLH115*), XP_016689171.2 (*GhFIT*), XP_040936509.1 (*GhPYE*). All accession numbers listed in Appendix A.

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
