# Peer review of "The Molecular Mechanism of GhbHLH121 in Response to Iron Deficiency in Cotton Seedlings"

_plants, 2023, doi:10.3390/plants12101955_

Round 1
Reviewer 1 Report
The authors have done a good job of investigating the Molecular Mechanism of GhbHLH121 in Response to Iron Deficiency in Cotton Seedlings. I have a few suggestions:
1. Cite the references in Figures 1, 8, or other, if you have borrowed the animated pictures from other sources.
2. Many gene names need to be italicized throughout the manuscript.
Author Response
Thanks for your evaluations and kind remind. Please see the attachment .
Reviewer 2 Report
The manuscript “The Molecular Mechanism of GhbHLH121 in Response to Iron Deficiency in Cotton Seedlings” described the functional analysis of the hub gene, GhbHLH121 in cotton in response to iron deficiency from several approaches including ectopic expression of GhbHLH121 in Arabidopsis, VIGS followed by chloroplast defect from microscopic analysis, protein-DNA, and protein-protein interaction to have a comprehensive understanding of the role of GhbHLH121 during the process of maintaining iron homeostasis in plants.
Major concerns
Did the authors perform any GUS reporter expression analysis to test the expression of the GhbHLH121 gene in the Arabidopsis’s plant tissue (root or shoot) in response to the Fe deficient and Fe well-acquired condition since the GUS expression vector was used as an ectopic expression vector?
Minor Errors
Page 4, Figure 1, Panel H,
Proteasonmal Degradation should be Proteasomal Degradation, spelling error;
Fe homeosotasis should be Fe homeostasis, spelling error;
Page 6, Page 199, VIGS was conducted, grammar error
Page 11, Line 338, LCI be should the abbreviation of Luciferase Complementation Imaging Assay
Figure 7, panel C and F
Effectors should be effector, grammar error;
Reporters 1 should be reporter 1, grammar error;
Page 16, Figure 8
Iron deficicency should be Iron deficiency, spelling error; same thing in supplemental figure S14
Author Response
Thanks for your kind suggestion. Please see the attachment.
Reviewer 3 Report
Dear Authors,
the VIGS approach for functional analyses of GhbHLH121 is not sufficiently described in paragraph 4.2. There are missing several important information, as a fragment size of amplified GhbHLH121 cloned into pTRV2 vector. Was the fragment cloned in antisense orientation or as a hairpin construct? If not, how you can support your observations about the suppression of GhbHLH121 by VIGS. Wouldn't this suppression be different depending on the type of construct? As you know, VIGS can be significantly improved throught virus-based expression of small inverted repeats in comparison to large sense or antisense cDNA transcripts. It would also be useful to add a picture of the construct for paragraph 4.3. Similarly, I consider paragraph 4.4 to be inadequately described, lacking information on amplified genes and housekeeing genes (internal control). I'm sorry, but for this reason, even part 2.3 of the results cannot be said to be sufficiently described. But this can be adjusted.
Author Response
Thanks for your kind suggestion. Please see the attachment
Reviewer 4 Report
Cotton is important for human and abiotic stress like saline-alkali soil limits its growth and yield. Alkaline stress hinders Fe uptake and causes iron deficiency in plants. The authors studied homologs of known factors involved in responses to iron deficiency and found GhbHLH121 plays a positive role in regulating cotton's responses. This manuscript is well written but still has several flaws.
Major:
Line 105: The -Fe condition is very extreme. Please design an experiment to show the iron deficienct condition used in this research mimics the natural alkline condition.
Line 133: The phylogenetic tree is too simple, please include more genes from relative species and statistic analysis.
Line 173: From the normalized heatmap, At gene expresses highly in roots and Dt gene expresses highly in seeds. Please explain how the heatmap was plotted and what the color legends stand for. This applies to all the heatmaps in this study. And please also explain why the authors concluded that roots and stems are where the bHLH genes express the most and their expression patterns are similar.
Line 180: The nucleus localization of At gene is very weak. Please reconsider the conclusion or include DAPI or other nuclear marker or use cell fractionation to confirm this.
Line 295: From the plant phenotype and heatmap, it seems like that bHLH121 alone could not activate downstream genes but the iron deficiency will activate bHLH121 and other downstream genes with bHLH121. Please explain this and add into discussion.
Line 330: This only shows the bHLH heterodimer resides in the nucleus, but not the free bHLH121 proteins. The author should conduct the co-localization assay to draw such conclusions.
Line 360: please explain why the authors tested the activation of bHLH121 by bHLH104 and bHLH115?
Line 416: This subsection just listed the results discoveried in this study. I didn’t see any discussion point in this paragraph.
Minor
Line 206: The significance should also be shown in the figures.
The manuscript is well written.
Author Response

(The authors gave the same response as above.)

Round 2
Reviewer 3 Report
Dear Authors,
thank you for your effort and time to explain the questions raised. I wish you much success in your future work.
Author Response
Thank you for your guidance
Reviewer 4 Report
I am satisfied with the majority of authors' responses except for the phylogenetic tree and the heatmap. Though genes in diploid cotton were added into the alignments, the phylogenetic tree still lacks numeric supports such as bootstrap. Please construct a new NJ tree with at least 1000 bootstraps or use other methods such as IQ-TREE. For Fig. 3H, I think this normalization was performed by "column" but not "row". Please confirm this. And since Ib genes and other Fe related genes were activated by -Fe, the differences in colors were not obvious amid the normalization. Please think about other methods.
And Line 450-458 in the main text is different from the authors' responses. For the GmbHLH104/115, the explanation should be included in the main text.
Author Response
Thanks for your suggestion. Please see the attachmen

Round 3
Reviewer 4 Report
I am satisfied with the current version.